# Macromolecular Brushes Based on Poly(L-Lactide) and Poly(ε-Caprolactone) Single and Double Macromonomers via ROMP. Synthesis, Characterization and Thermal Properties

**DOI:** 10.3390/polym11101606

**Published:** 2019-10-01

**Authors:** Christiana Nikovia, Eleftheria Sougioltzoupoulou, Vyron Rigas, Marinos Pitsikalis

**Affiliations:** Department of Chemistry, Industrial Chemistry Laboratory, National and Kapodistrian University of Athens, Panepistimiopolis Zografou, 15771 Athens, Greece; xristiana2309@gmail.com (C.N.); sou.eleftheria@gmail.com (E.S.); vyron.rigas@gmail.com (V.R.)

**Keywords:** ring opening polymerization (ROP), ring opening metathesis polymerization (ROMP), norbornene, copolymerization, macromonomer, polymer brushes, thermogravimetric analysis (TGA), differential scanning calorimetry (DSC)

## Abstract

Single and double poly(L-lactide) (PLLA) and poly(ε-caprolactone) (PCL) macromonomers having a norbornenyl polymerizable group were prepared by conventional Ring Opening Polymerization (ROP). These macromonomers were further subjected to ring opening metathesis polymerization (ROMP) reactions in order to produce double polymer brushes consisting of PLLA or PCL side chains on a polynorbornene (PNBE) backbone. Statistical or block ring opening metathesis copolymerization of the PLLA and PCL macromonomers afforded the corresponding random and block double brushes. Sequential ROMP of the single PLLA, PCL and PLLA macromonomers resulted in the synthesis of the corresponding triblock copolymer brush. The molecular characteristics of the macromolecular brushes were obtained by ^1^H-NMR spectroscopy and Size Exclusion Chromatography. The thermal properties of the samples were studied by thermogravimetric analysis, TGA, Differential Thermogravimetry, DTG and Differential Scanning Calorimetry, DSC.

## 1. Introduction

Macromolecular engineering has been proven to be the tool for the manipulation of polymeric materials properties in recent years [1,2,3]. So far, the copolymer composition, the constituent block molecular weights and the chemical nature of these blocks have been the factors, which are responsible for determining the physical behavior of the macromolecules and their applications [4,5]. However, the recent tremendous advances in Polymer Chemistry has allowed new polymerization methods to emerge providing an opportunity to control molecular characteristics and the microstructure and in addition to expand to new monomers that have never been polymerized in a controlled way. Among these techniques, recent progress in controlled radical, cationic and coordination polymerization has to be mentioned [6,7,8,9,10,11,12,13,14]. Furthermore, the application of characteristic organic chemistry reactions, e.g., click chemistry [15] and Suzuki coupling [16], the synthesis of novel initiators, linking agents, chain transfer agents, organocatalytic species and metal complexes have allowed the transition of Polymer Chemistry to a new era leading to the synthesis of complex macromolecular architectures in an elegant and efficient way [17,18].

Among these effective and relatively new polymerization techniques is definitely Ring Opening Metathesis Polymerization (ROMP) [19]. It is based on the classical olefin metathesis reaction. However, the synthesis of novel catalysts, working in a more or less “living” fashion has offered tremendous opportunities for the synthesis of novel polymeric materials. Ru-, Mo- and W-based catalysts allowed functional group tolerance, and provided high catalytic efficiency leading to products with controlled molecular characteristics and stereochemistry. The combination of ROMP with other polymerization techniques has allowed for the synthesis of novel, non-linear, complex polymeric architectures [20,21,22,23,24].

During the past few years the synthesis of branched materials has been a pivotal area of research. Parameters such as the composition, the number of grafted chains on the backbone, the relative molecular weights of the backbone and the side chains along with the grafting density are able to influence the bulk and solution properties of these materials. The grafting “onto”, grafting “from” and grafting “through” methodologies have been developed in the past and have been efficiently employed, in combination with various polymerization techniques, for the synthesis of a huge variety of macromolecular branched structures [25,26,27,28,29,30,31,32,33,34,35,36,37].

There is an ongoing effort in our group towards the combination of the grafting “through” or “macromonomer” technique with ROMP and other polymerization methods for the synthesis of branched structures and especially polymer brushes or polymacromonomers. Their constrained geometry and topology is responsible for the unique properties of these materials both in solution and in bulk. Along these lines, there are anionic, atom transfer radical, ring opening and coordination polymerization techniques that have been employed for the synthesis of macromonomers bearing a norbornenyl end-group. These macromonomers have been either homopolymerized or copolymerized both statistically or by sequential macromonomer addition to afford a variety of brushes. Other more complex structures, such as brushes on brushes have also been synthesized. The materials have been thoroughly characterized and their thermal properties have been investigated [38,39,40,41,42].

In a previous study, we reported the synthesis of norbornene-functionalized poly(L-lactide), PLLA, and poly(ε-caprolactone), PCL, macromonomers from stannous octoate-catalysed ring opening polymerization (ROP), in the presence of a mono-hydroxylated norbornene derivative as an initiator. Subsequent ROMP of these PLLA and PCL macromonomers using the 1st generation ruthenium-based Grubbs catalyst afforded narrowly dispersed brush copolymers with different side chains [39]. The PLLA and PCL macromonomers were combined in order to prepare two different series of brushes including brush block copolymers and brush statistical copolymers.

The present study is a step forward in our efforts to manipulate the macromolecular architecture. Employing single PLLA and PCL macromonomers, the synthesis of a triblock copolymer brush was attempted in order to check the level of control of ROMP for the synthesis of complex structures. In addition, the synthesis of novel polymer brushes based on double PLLA and PCL macromonomers were described along with their thermal decomposition behavior, studied by thermogravimetric analysis (TGA), differential thermogravimetry (DTG) and Differential Scanning Calorimetry (DSC).

## 2. Materials and Methods

### 2.1. Materials

High vacuum and/or Schlenk techniques were applied in all purification and synthetic steps [43,44,45,46]. L-lactide (LLA) was recrystallized in acetone and subsequently dried overnight under vacuum and stored in the glovebox, whereas ε-caprolactone (CL) was dried and vacuum distilled from calcium hydride twice, prior to use. Stannous octoate [Sn(Oct)_2_], [RuCl_2_(=CHPh)(PCy_3_)_2_] (1st generation Grubbs catalyst), 5-norbornene-2-methanol 98% (mixture of *endo* and *exo)* and 5-norbornene-2,3-dimethanol were purchased from Aldrich and used without further manipulation. The 3rd generation Grubbs catalyst was prepared upon treatment of the 2nd generation Grubbs catalyst (Aldrich) with 3-bromopyridine, following literature conditions [47]. Toluene was dried over calcium hydride, degassed and distilled to calibrated cylinder containing oligostyryllithium. Dichloromethane was distilled from calcium hydride and dried over molecular sieves (3 Å or 4 Å) and distilled prior to use. Ethyl vinyl ether and methanol were used as received.

### 2.2. Characterization Techniques

SEC (Size Exclusion Chromatopgraphy) experiments were carried out using a modular instrument consisting of a Waters Model 510 pump, a Waters Model U6K sample injector, a Waters Model 401 differential refractometer (Waters Corporation, Milford, MA, USA), and a set of three μ-styragel columns with a continuous porosity range from 10^6^ to 10^3^ Å. The columns were housed in an oven thermostatted at 40 °C. THF or CHCl_3_ was the carrier solvent at a flow rate of 1 mL min^−1^. The instrument was calibrated with polystyrene standards covering the molecular weight range from 3000 up to 600,000. ^1^H NMR spectra were recorded in chloroform-d at 30 °C with a Varian Unity Plus 300/54 NMR spectrometer (Varian, Inc., Palo Alto, CA, USA). Thermogravimetric analysis (TGA) experiments were carried out using a Q50 Model of TA Instruments (New Castle, DE, USA) employing samples of approximately 10 mg. The heating rate was adjusted to 10 °C min^−1^. Differential Scanning Calorimetry (DSC) experiments were conducted using a 2910 modulated DSC model from TA Instruments. The samples were heated at a rate of 10 °C/min from −30 to 180 °C. The second heating results were obtained in all cases.

### 2.3. Synthesis of NBE-PCL Macromonomer

Macromonomers were synthesized via ROP of CL using 5-norbornene-2-methanol (mixture of isomers) as the initiator and Sn(Oct)_2_ as the catalyst. In a typical experiment 10 mL of CL (9 × 10^−2^ mol) was distilled from CaH_2_ into a flame dried 100 mL schlenk flask equipped with a magnetic stirring bar and degassed via three freeze-pump-thaw cycles. 30 mL of toluene was added under argon atmosphere to the schlenk flask using a syringe in order to dissolve the monomer. Then, the initiator (0.25 mL, 2 × 10^−3^ mol) and the catalyst Sn(Oct)_2_ (0.2 mL, 6 × 10^−4^ mol) were injected into the monomer solution under an argon atmosphere and the reaction flask was immersed in a thermostatted oil bath at 120 °C and stirred for 24 h. After cooling to room temperature, the crude product was precipitated into cold methanol (400 mL). The precipitate was isolated by filtration, washed with methanol and dried in a vacuum oven overnight.

^1^HNMR (CDCl_3_): δ(ppm) 6.14–5.94 (2H, olefinic protons on the norbornene ring) 4.07–4.05 (m, CH_2_OCO on PCL) 3.69–3.62 (m, PCL-CH_2_OH) 2.83 (m, CH on the norbornene ring) 2.68 (m, CH on the norbornene ring) 2.32–2.28 (m, OCOCH_2_ on PCL) 1.67–1.60 (m, CH_2_ on PCL) 1.42–1.32 (m, CH_2_ on PCL) 0.56 (CH_2_ on the norbornene ring)

### 2.4. Synthesis of NBE-(PCL)_2_ Double Macromonomer

A similar approach was adopted previously, except that 5-norbornene-2,3-dimethanol was used as the initiator instead of 5-norbornene-2-methanol. Two samples, NBE-(PCL)_2_ #1 and NBE-(PCL)_2_ #2, were synthesized. Typical quantities for the synthesis of NBE-(PCL)_2_ #2 (Table 1) are the following: 5-norbornene-2,3-dimethanol 0.32 g, CL 10 mL (9 × 10^−2^ mol), Sn(Oct)_2_ 0.2 mL (6 × 10^−4^ mol) and toluene 30 mL.

### 2.5. Synthesis of NBE-PLLA Macromonomer

An oven dried 100 mL schlenk flask equipped with a magnetic stir bar was charged with 10 g of recrystallized LLA. The flask was thoroughly degassed to remove traces of water and acetone from the monomer. The desired amount of degassed anhydrous toluene (50 mL) was added via syringe under an argon atmosphere to dissolve the monomer. Then, the initiator (0.25 mL, 2 × 10^−3^ mol) and Sn(Oct)_2_ (0.2 mL, 6 × 10^−4^ mol) were injected to the monomer solution under argon atmosphere and the reaction flask was immersed in a thermostatted oil bath at 120 °C. After 24 h, the content was cooled to room temperature and the polymer was precipitated into methanol. The macromonomer was isolated by filtration, washed with methanol and dried in a vacuum oven overnight.

^1^H NMR (CDCl_3_): δ (ppm) 6.16–5.92 (2H, olefinic protons on the norbornene ring) 5.20–5.13 (m, CH on PLA) 4.39–4.32 (m, -CH_2_OC(O)- of NBE group) 2.83 (s,1H, allylic proton of NBE group) 2.66 (s,1H, allylic proton of NBE group) 1.60–1.44 (br, -CH_3_ of PLA backbone) 1.37–1.24 (m, -CH< and >CHCH_2_CH< of NBE group).

### 2.6. Synthesis of NBE-(PLLA)_2_ Double Macromonomer

A similar approach was adopted previously, except that 5-norbornene-2,3-dimethanol was used as the initiator instead of 5-norbornene-2-methanol. Two double macromonomers were synthesized, NBE-(PLLA)_2_ #1 and NBE-(PLLA)_2_ #2. Typical quantities for the synthesis of NBE-(PLLA)_2_ #2 are the following: 5-norbornene-2,3-dimethanol 0.2 g, LLA 5.5 g, Sn(Oct)_2_ 0.2 mL and toluene 40 mL.

### 2.7. Synthesis of Double Brushes via Ring Opening Metathesis Polymerization of NBE-(PLLA)_2_ and NBE-(PCL)_2_ Macromonomers

An oven-dried schlenk flask was charged with 1g of NBE-(PLLA)_2_ #1 macromonomer and a stir bar. The schlenk was evacuated and backfilled with argon three times. The desired amount of degassed anhydrous dichloromethane (8 mL) was added via a syringe under an argon atmosphere to dissolve the macromonomer. A stock solution of the 3rd generation Grubbs Ru catalyst (6 mg) in degassed anhydrous CH_2_Cl_2_ (5 mL) was prepared in a separate schlenk flask. The macromonomer solution was injected into the catalyst solution. The reaction was allowed to run at room temperature for 3 h. The polymerization was terminated by the addition of 1 mL of ethyl vinyl ether and stirred for an additional 30 min. The reaction mixture was then poured into excess cold methanol with stirring and the precipitates were isolated by filtration, washed with methanol and dried in a vacuum oven overnight to yield a white solid. The exact same procedure was followed for the homopolymerization of the NBE-(PCL)_2_ double macromonomer.

### 2.8. Synthesis of Double Brushes Block Copolymer via Ring Opening Metathesis Polymerization of NBE-(PLLA)_2_ and NBE-(PCL)_2_ Macromonomers

An oven-dried schlenk flask was charged with the desired amount of NBE-(PLLA)_2_ #1 macromonomer (0.5 g) for the first block and a stir bar. The schlenk flask was evacuated and backfilled with argon three times. The desired amount of degassed anhydrous dichloromethane (10 mL) was added via a syringe under an argon atmosphere to dissolve the macromonomer. A stock solution of the 1st generation Grubbs Ru catalyst (20 mg, 2.4 × 10^−5^ mol) in degassed anhydrous CH_2_Cl_2_ (5 mL) was prepared in a separate schlenk flask. The macromonomer solution was injected into the catalyst solution. The reaction was allowed to run at room temperature for 4 h. After the first polymerization was completed, the desired amount of NBE-(PCL)_2_ #1 macromonomer (0.5 g) was added as a solution in CH_2_Cl_2_ (5 mL). After 24 h the polymerization was terminated by the addition of 1 mL of ethyl vinyl ether and stirred for an additional 30 min. The reaction mixture was then poured into excess cold methanol with stirring and the precipitates were isolated by filtration, washed with methanol and dried in a vacuum oven overnight to yield a white solid.

### 2.9. Synthesis of Double Brushes Statistical Copolymer via Ring Opening Metathesis Polymerization of NBE-(PLLA)_2_ and NBE-(PCL)_2_ Macromonomers

The statistical double brush copolymer was synthesized using a similar procedure as the block copolymer except that the two types of macromonomers were added simultaneously to the same schlenk flask and stirred to ensure homogeneous mixing before injection into the catalyst solution.

### 2.10. Synthesis of Brush Triblock Copolymer via Ring Opening Metathesis Polymerization of NBE-PLLA and NBE-PCL Macromonomers

An oven-dried schlenk flask was charged with 0.5 g of NBE-PLLA macromonomer for the first block and a stir bar. The schlenk was evacuated and backfilled with argon three times. The desired amount of degassed anhydrous dichloromethane (10 mL) was added via a syringe under argon atmosphere to dissolve the macromonomer. A stock solution of the 1st generation Grubbs Ru catalyst (20 mg, 2.4 × 10^−5^ mol) in degassed anhydrous CH_2_Cl_2_ (5 mL) was prepared in a separate Schlenk flask. The macromonomer solution was injected into the catalyst solution. The reaction was allowed to run at room temperature for 4 h. After the first polymerization was completed, 0.5 g of NBE-PCL macromonomer was added as a solution in CH_2_Cl_2_ (5 mL). The polymerization of the second block was allowed to proceed for 4 h and then a new amount of NBE-PLLA macromonomer (0.5 g dissolved in 5 mL of CH_2_Cl_2_) was added. After 24 h the polymerization was terminated by the addition of 1 mL of ethyl vinyl ether and stirred for an additional 30 min. The reaction mixture was then poured into excess cold methanol with stirring and the precipitates were isolated by filtration, washed with methanol and dried in a vacuum oven overnight to yield a white solid.

## 3. Results

### 3.1. Synthesis of Single and Double Macromonomers

The synthesis and the detailed molecular characterization of the single and double macromonomers have been described in a previous publication [39]. Additional double macromonomers were included in this study. Well-defined structures were obtained, as shown by SEC and NMR analysis. The molecular characteristics of the macromonomers are given in Table 1.

The synthesis of the macromonomers were based on the ROP of LLA and CL employing either 5-norbornene-2-methanol or 5-norbornene-2,3-dimethanol as initiators in the presence of Sn(Oct)_2_, as the catalyst, according to the following reactions, given in Figure 1 and Figure 2:

### 3.2. Synthesis of Homopolymer Double Brushes

The double macromonomers NBE-(PLLA)_2_ #1 and NBE-(PCL)_2_ #1 were homopolymerized in CH_2_Cl_2_ solutions using the Grubbs 3rd generation catalyst. In previous studies, [48,49,50] it has been shown that this catalyst is efficient for the polymerization of 2,3-disubstituted norbornenes. In addition, this catalyst has a high initiation rate and leads to polymeric products with narrow molecular weight distributions [47,48,50]. The polymerization reactions are given in Figure 3 and Figure 4, whereas the molecular characteristics of the double brushes are provided in Table 2. The samples are denoted as DB-PLLA and DB-PCL, respectively.

Upon the addition of the macromonomer solution to the originally green solution of the catalyst, the colour immediately changed to yellow indicating the high initiation rate of the catalyst. The polymerization was monitored by SEC and the corresponding traces are provided in Figure 5 and Figure 6. In the case of the homopolymerization of the NBE-(PLLA)_2_ #1 macromonomer it was found that the double polymer brush had a narrow molecular weight distribution and the macromonomer was almost quantitatively consumed during the polymerization reaction. However, a small trace of unreacted macromonomer (about 10%) was found by SEC analysis. This result shows that the steric hindrance for the polymerization of the double macromonomer was very pronounced and that maybe extended polymerization times are required for the complete consumption of the macromonomer. In the case of the homopolymerization of the NBE-(PCL)_2_ #1 macromonomer the reaction was extended for one more hour and the SEC analysis revealed quantitative consumption of the macromonomer. However, in this case the molecular weight distribution was broader, probably indicating that the sample had a higher degree of chemical heterogeneity, or in other words that the product was a mixture of brushes with different degrees of polymerization. In the literature, there are a few examples concerning the polymerization of double macromonomers [30,51,52]. In most cases the polymerization procedures suffered similar problems. Broader molecular weight distributions and higher amounts of unreacted macromonomers were obtained compared to our case.

The formation of the desired products was also monitored by ^1^H NMR spectroscopy, where the characteristic signals of the side chains and the backbone were obvious (Figure 7 and Figure 8).

### 3.3. Synthesis of Statistical and Block Copolymeric Brushes via Ring Opening Metathesis Polymerization of the Double Macromonomers

Simultaneous copolymerization of the NBE-(PLLA)_2_ #1 and NBE-(PCL)_2_ #1 double macromonomers afforded the corresponding double copolymer brush, as shown in Figure 9. The sample is denoted as SDBC. The copolymerization was conducted in the presence of the inexpensive and effective in polymerization reactions Grubbs 1^st^ generation catalyst at room temperature. The reaction was monitored by SEC and ^1^H NMR spectroscopy (Figure 10 and Appendix A, respectively). It is important to note that under these experimental conditions the consumption of the double macromonomers was quantitative and that the final product had a narrow molecular weight distribution. Therefore, well-defined products can be obtained through this approach overcoming the steric hindrance effects, which are associated with the nature of the double macromonomers, despite the problems that have been previously reported in the literature in similar cases. The molecular characteristics of the sample are provided in Table 3.

The double brush block copolymacromonomer was also synthesized by sequential polymerization of the respective double macromonomers. Initially, the NBE-(PLLA)_2_ #1 was homopolymerized, followed by the addition of the NBE-(PCL)_2_ #1 macromonomer, as shown in Figure 11. The sample is denoted as BDBC. A previous study regarding the block copolymerization of the single macromonomers revealed that the order of addition of the macromonomers does not play a crucial role in the copolymerization procedure, since the polymerizable group is the same for both macromonomers [39]. The block copolymerization was conducted under similar experimental conditions as in the case of the statistical double brush. SEC (Figure 12) and ^1^H NMR (Appendix A) analysis confirmed the synthesis of the desired product in a very well controlled fashion. The narrow molecular weight distribution of the final product and the composition, which is very close to the stoichiometric values, unambiguously indicates that this procedure efficiently leads to the synthesis of complex brush-like architectures. The molecular weights of the double macromonomer brush copolymer are given in Table 3.

### 3.4. Synthesis of Triblock Copolymer Brush via Ring Opening Metathesis Polymerization of Single Macromonomers

The synthesis of a series of well-defined block copolymacromonomers by sequential polymerization of NBE-PLLA and NBE-PCL macromonomers was previously reported [20]. In order to further confirm the controlled nature of the reaction scheme the synthesis of a triblock copolymer brush was attempted by the sequential polymerization of single macromonomers, as shown in Figure 13. The reaction was monitored by SEC and ^1^H NMR spectroscopy. The data are given in Figure 14 and Appendix A, whereas the molecular characteristics of the sample are listed in Table 3. The sample is denoted as B-(PLLA-b-PCL-b-PLLA). The SEC trace of the final product indicated the presence of a small amount (less than 10%) of unreacted NBE-PLLA macromonomer. ROMP is a well-controlled polymerization technique. However, it is susceptible to various side reactions leading to lower reactivities and to termination or transfer reactions [53]. The addition of further polymerization steps and the extension of polymerization time allows these side reactions to proceed leading to several byproducts. Nevertheless, these side events were minimized under the adopted experimental conditions, thus achieving the best control over the macromolecular architecture and giving the possibility to prepare even more complex structures.

### 3.5. Thermal Decomposition of the Macromonomers and the Polymer Brushes

The thermal stability of the polymer brushes were evaluated by TGA and DTG measurements, which has been incorporated in Table 4. Characteristic DTG plots for the macromonomers and the brushes are given in Figure 15 and Figure 16. More data has been provided at the Appendix A, SIS (Appendix A). As reported in the literature for the respective PLLA and PCL homopolymers, the later is more thermally stable, whereas both present a single decomposition curve, indicative of a rather simple thermal decomposition mechanism [54]. Similar results were obtained for both the single and the double macromonomers revealing that the presence and the position of the norbornenyl group did not affect the thermal decomposition profile of the PLLA and PCL macromonomers.

TGA and DTG data on the polymer brushes coming from both the single and the double macromonomers showed similar behavior, without any specific effect of the exact arrangement of the PLLA and PCL side chains. The polymer brushes synthesized via the homopolymerization of the respective NBE-(PLLA)_2_ #1 and NBE-(PCL)_2_ #1 double macromonomers showed DTG plots with two well-resolved peaks. The major peak at a lower temperature corresponded to the thermal decomposition of the PLLA or PCL side chains and the minor one at a much higher temperature corresponded to the thermal decomposition of the PNBE backbone of the polymer brush. Similar behavior was obtained for the statistical and block brushes, produced by the double macromonomers and the triblock copolymer brush, produced by the single macromonomers. For these samples three distinctive thermal decomposition peaks were observed corresponding to the respective PLLA and PCL side chains and the PNBE backbone. Therefore, each polymeric chemical species is thermally decomposed separately, first the PLLA side chains, then the PCL side chains and finally the PNBE backbone. Similar results have been reported previously regarding the thermal decomposition of the statistical and block copolymer brushes prepared by the ROMP of single NBE-PLLA and NBE-PCL macromonomers [39].

### 3.6. Thermal Transitions of the Macromonomers and the Polymer Brushes by Differential Scanning Calorimetry

The thermal transitions of the macromonomers and the prepared brushes were studied by Differential Scanning Calorimetry (DSC). The results are listed in Table 5, whereas representative thermograms are given in the SIS (Appendix A).

The single and double macromonomers based on PCL show the characteristic melting peak of PCL. The double macromonomer has lower Tm and ΔHm values revealing that the change in the architecture reduced the crystallinity of the polymer. The degree of crystallinity was calculated from the ratio ΔHm/(ΔHm)_∞_, where ΔHm was the enthalpy of melting of the specific sample and (ΔHm)_∞_ the enthalpy of melting of the ideal crystal with 100% crystallinity. For PCL (ΔHm)_∞_ = 139.5 J/g [55]. In the case of the DB-PCL the crystallinity was further reduced along with the Tm value, due to the frustration induced by the macromolecular architecture, which restricted the formation of well- organized crystalline regions.

In the case of the NBE-PLLA macromonomer, an exothermic peak was observed due to cold crystallization followed by a rather broad melting transition. Changing the architecture to the double macromonomer, the cold crystallization was no longer observed upon heating at 10 °C/min. However, two melting peaks were obvious. It is well documented that PLLA has the ability to form four different crystal phases called α, β, γ, and α’ (α’ is the disordered form of α) [56,57]. Therefore, the Tm_2_ and Tm_3_ values corresponded to the melting of the α and the α’ phases that coexisted in the solid state. The overall degree of crystallinity substantially reduced for the double macromonomer, revealing that the effect of architecture was even more pronounced in the case of PLLA than in PCL. For the calculation of the degree of crystallinity for PLLA, the value (ΔHm)_∞_ = 135 J/g was employed [58].

The DB-PLLA showed a broad melting transition at 138 °C and the degree of crystallinity was also very low as was the case for the constituting double macromonomer.

The double brushes block copolymer BDBC showed two different melting peaks corresponding to the PCL and the PLLA crystal phases, respectively. Due to the constrained structure, the degree of crystallization was restricted, especially for the PLLA crystal phases. Finally, the triblock copolymer brush revealed the melting of the PCL crystal phases and the double melting peaks of the PLLA crystal phases. The degree of crystallization was also reduced in this case. However, it was not so pronounced, as in the case of the structures produced from the double macromonomers.

## 4. Conclusions

Well-defined single and double poly(L-lactide) and poly(ε-caprolactone) macromonomers having a norbornenyl group were synthesized via conventional Ring Opening Polymerization. The double macromonomers were then homopolymerized via ring opening metathesis polymerization reactions in order to produce double polymer brushes consisting of PLLA or PCL side chains on a polynorbornene backbone. Simultaneous copolymerization of the double macromonomers afforded the statistical double copolymer brush, whereas sequential addition of the respective macromonomers allowed for the synthesis of the block double macromonomer brush. Sequential ROMP of the single PLLA, PCL and PLLA macromonomers resulted in the synthesis of the corresponding triblock brush. Well-defined products were obtained in all cases as was revealed by ^1^H-NMR spectroscopy and Size Exclusion Chromatography. The thermal properties of the samples were studied by thermogravimetric analysis. It was shown that the thermal decomposition pattern was not affected by the macromolecular architecture, since each polymeric species was independently decomposed at a specific range of temperatures. DSC analysis, however, revealed that the complex macromolecular architecture prevents the crystallization of the PCL and PLLA blocks. This effect is even more pronounced in the case of the structures produced by double macromonomers.

## Figures and Tables

**Figure 1 polymers-11-01606-f001:**
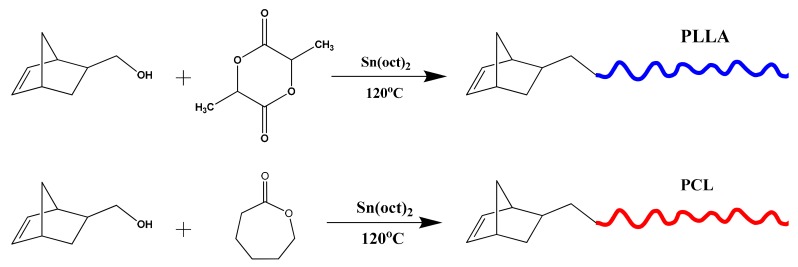
Synthesis of NBE-PLLA and NBE-PCL macromonomers.

**Figure 2 polymers-11-01606-f002:**
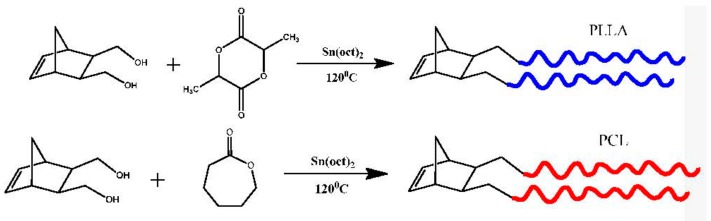
Synthesis of NBE-(PLLA)_2_ and NBE-(PCL)_2_ double macromonomers.

**Figure 3 polymers-11-01606-f003:**
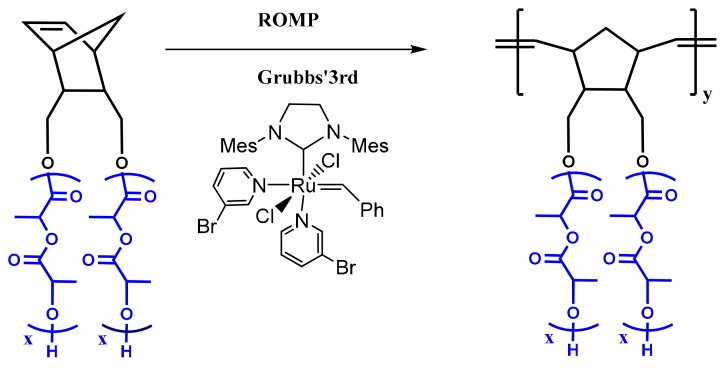
Synthesis of DB-PLLA.

**Figure 4 polymers-11-01606-f004:**
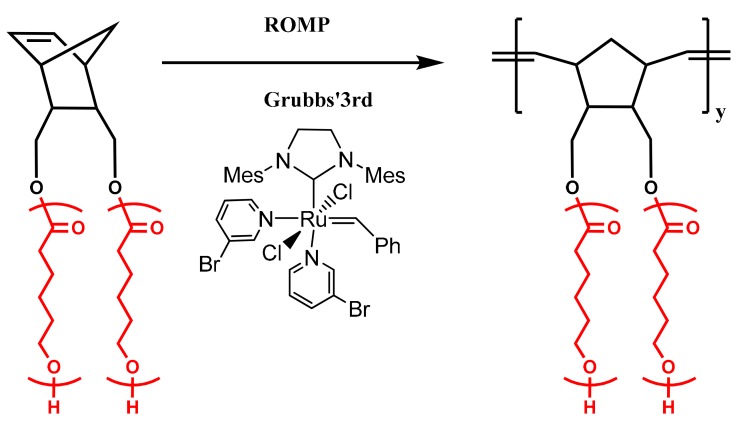
Synthesis of DB-PCL.

**Figure 5 polymers-11-01606-f005:**
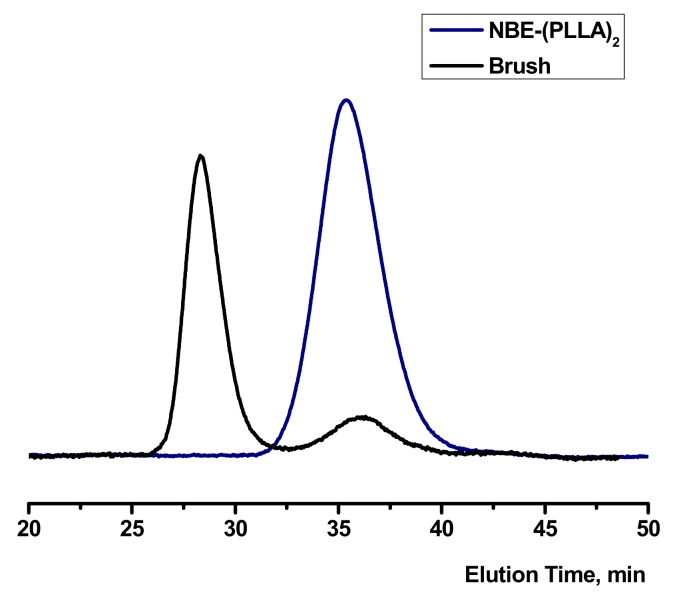
SEC traces monitoring the synthesis of DB-PLLA.

**Figure 6 polymers-11-01606-f006:**
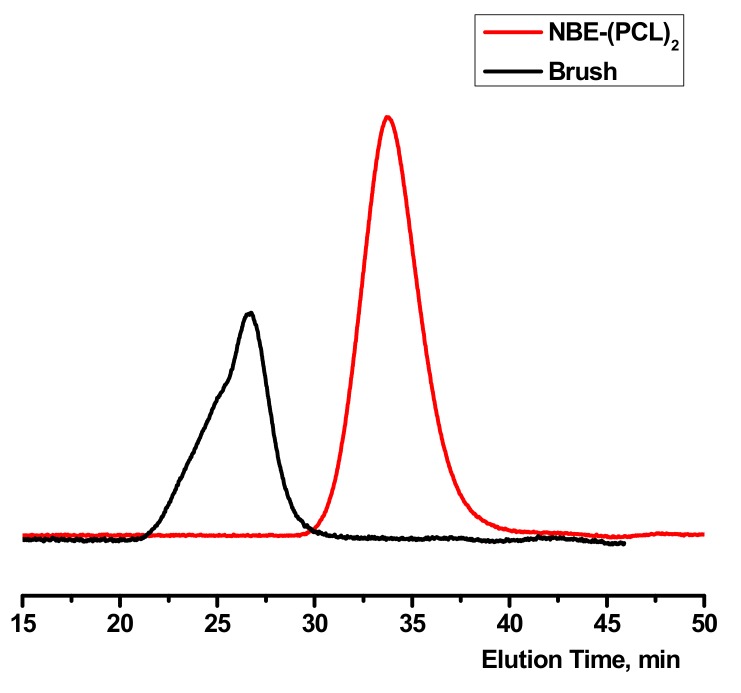
SEC traces monitoring the synthesis of DB-PCL.

**Figure 7 polymers-11-01606-f007:**
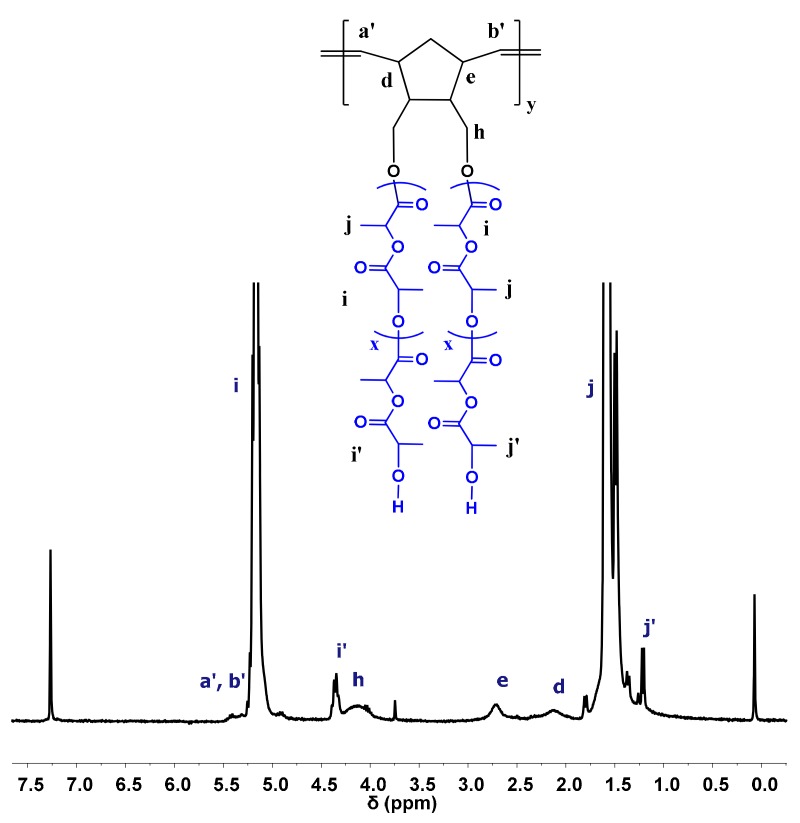
^1^H NMR spectrum of sample DB-PLLA in CDCl_3_.

**Figure 8 polymers-11-01606-f008:**
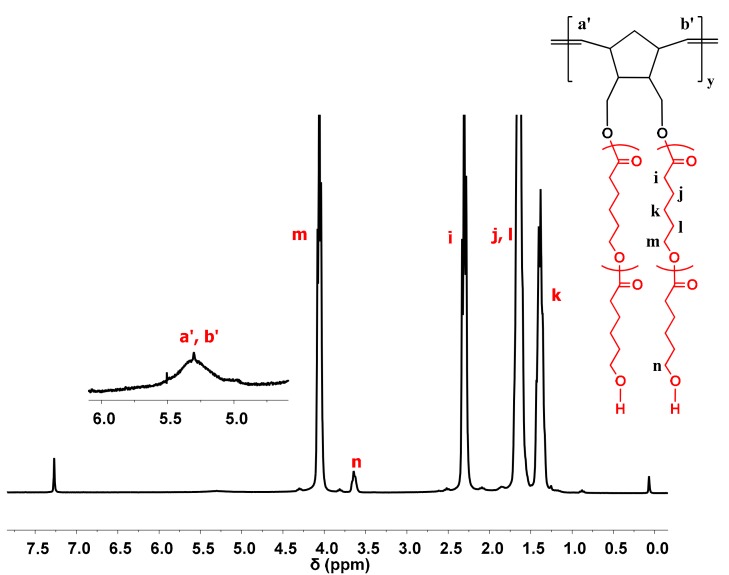
^1^H NMR spectrum of sample DB-PCL in CDCl_3_.

**Figure 9 polymers-11-01606-f009:**
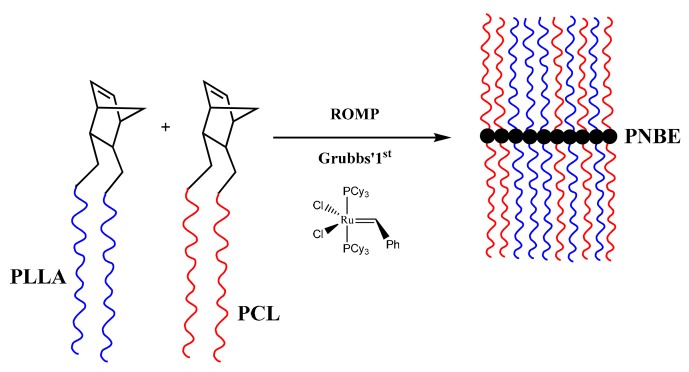
Synthesis of SDBC.

**Figure 10 polymers-11-01606-f010:**
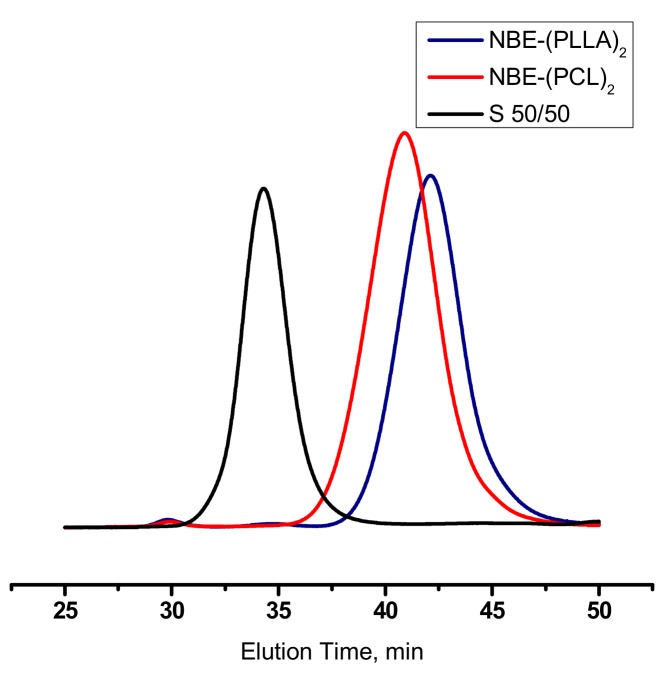
SEC traces monitoring the synthesis of SDBC.

**Figure 11 polymers-11-01606-f011:**
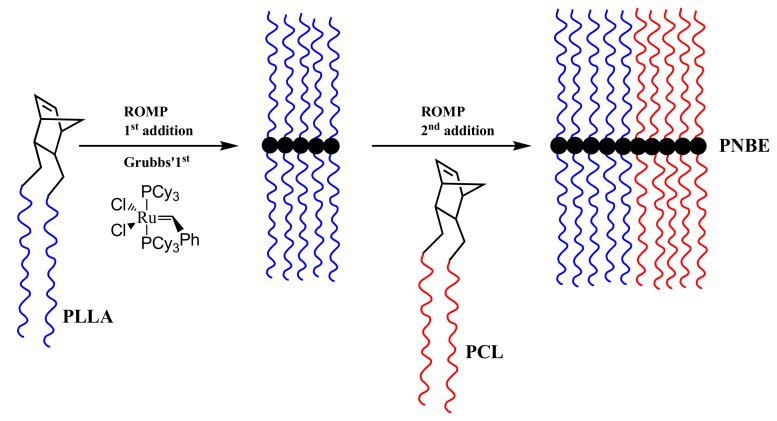
Synthesis of BDBC.

**Figure 12 polymers-11-01606-f012:**
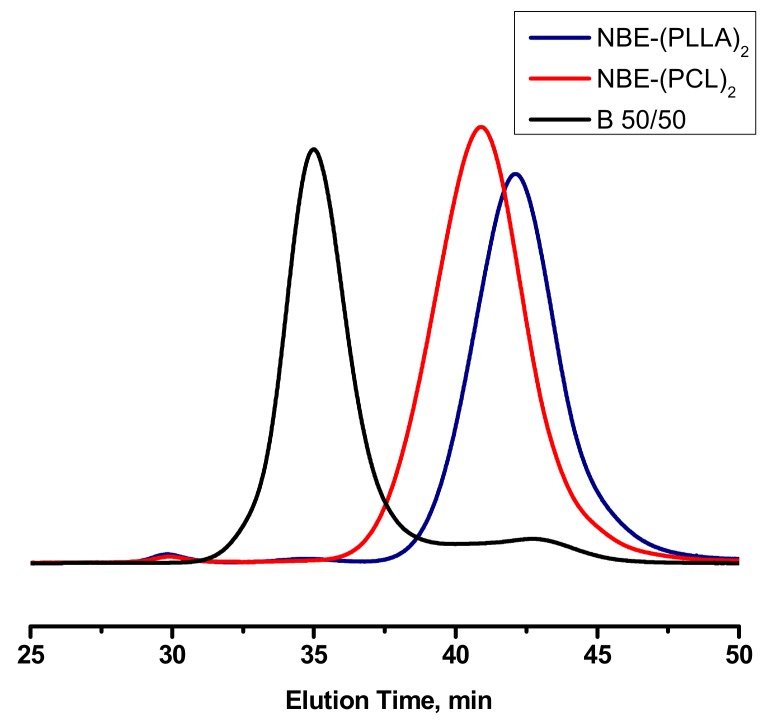
SEC traces monitoring the synthesis of BDBC.

**Figure 13 polymers-11-01606-f013:**
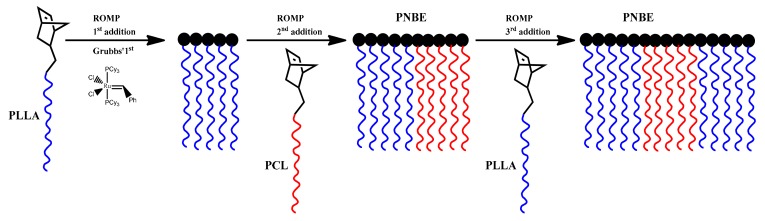
Synthesis of B-(PLLA-b-PCL-b-PLLA).

**Figure 14 polymers-11-01606-f014:**
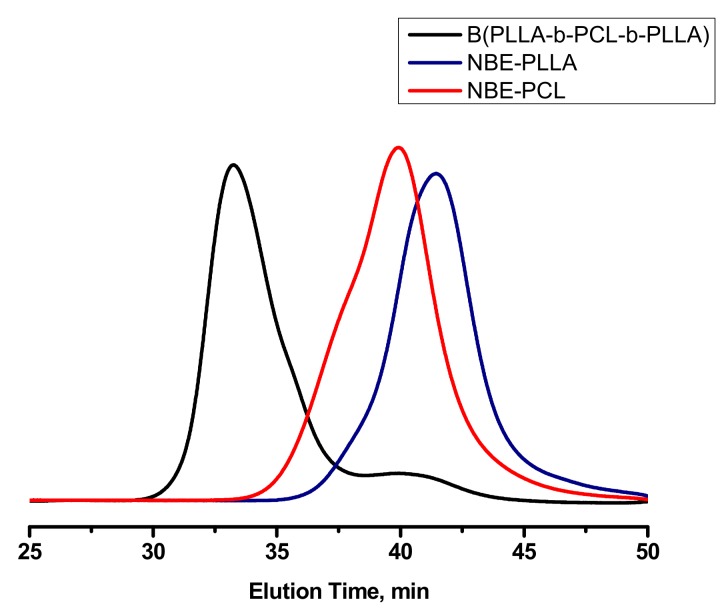
SEC traces monitoring the synthesis of B-(PLLA-b-PCL-b-PLLA).

**Figure 15 polymers-11-01606-f015:**
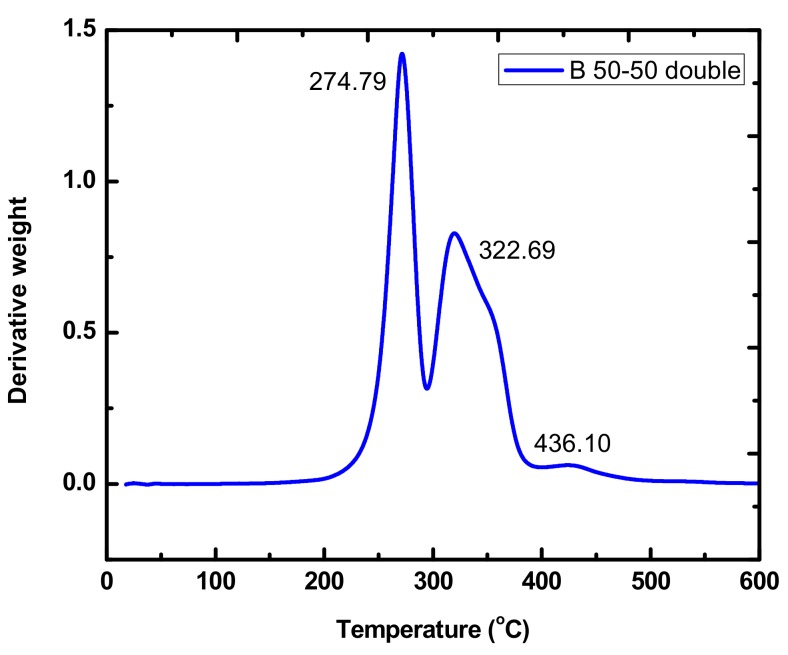
DTG plot of the sample BDBC.

**Figure 16 polymers-11-01606-f016:**
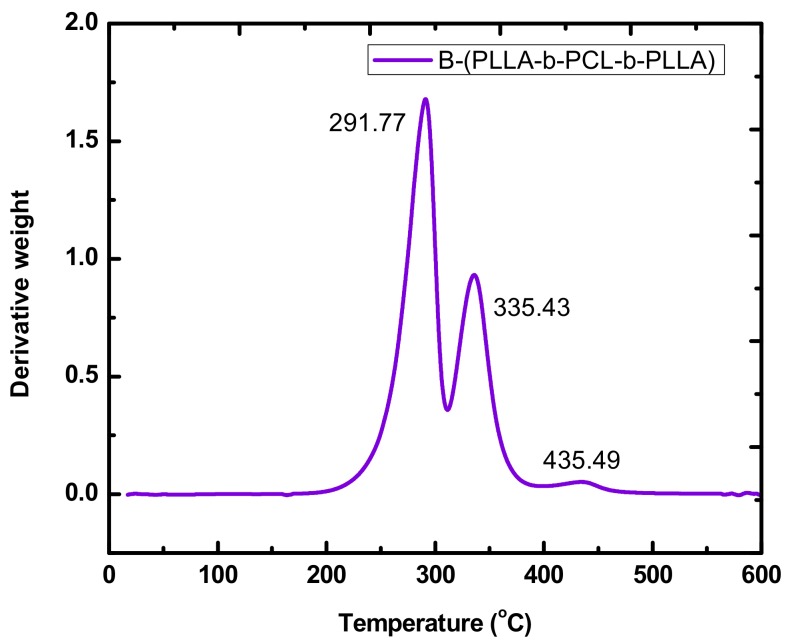
DTG plot of the sample B-(PLLA-b-PCL-b-PLLA).

**Table 1 polymers-11-01606-t001:** Molecular characteristics of the norbornenyl macromonomers.

Macromonomers	M_w, SEC_ ^b^ (g mol ^−1^)	M_w_/M_n_ ^b^	M_n, NMR_ ^c^ (g mol ^−1^)
NBE-PLLA ^a^	5800	1.18	4200
NBE-PCL ^a^	11,400	1.32	12,400
NBE-(PLLA)_2_ #1 ^a^	4200	1.30	2900
NBE-(PLLA)_2_ #2 ^a^	9000	1.17	4270
NBE-(PCL)_2_ #1 ^a^	6480	1.30	4550
NBE-(PCL)_2_ #2 ^a^	11,300	1.13	8500

^a^ Conditions: T = 120 °C/solvent: toluene_/_reaction time: 24 h ^b^ by SEC in THF, calibrated with linear polystyrene standards ^c^ by ^1^NMR.

**Table 2 polymers-11-01606-t002:** Molecular characteristics of the double homopolymer brushes.

Sample	M_w_ ^a^	M_w_/M_n_ ^a^	f ^b^
DB-PLLA ^c^	36,900	1.10	9
DB-PCL ^d^	116,350	1.40	18

^a^ by SEC in THF at 40 °C. ^b^ number of branches ^c^ the macromonomer NBE-(PLLA)_2_ #1 that was employed ^d^ the macromonomer NBE-(PCL)_2_ #1 that was employed.

**Table 3 polymers-11-01606-t003:** Molecular characteristics of the statistical and block copolymeric brushes along with the triblock copolymer brush via Ring Opening Metathesis Polymerization of the single macromonomers.

Sample	M_w_ ^a^	M_w_/M_n_ ^a^	% PLLA (*w*/*w*) ^b^	f ^c^
SDBC ^d^	47,000	1.13	61	7/3
BDBC ^e^	37,900	1.13	62	6/3
B-(PLLA-b-PCL-b-PLLA)	60,300	1.17	76	8/2

^a^ by SEC in THF at 40 °C. ^b^ by ^1^H NMR in CDCl_3_ at room temperature. ^c^ number of branches (PLLA branches/PCL branches). ^d^ statistical copolymerization of NBE-(PLLA)_2_ #1 and NBE-(PCL)_2_ #1. ^e^ block copolymerization of NBE-(PLLA)_2_ #1 and NBE-(PCL)_2_ #1.

**Table 4 polymers-11-01606-t004:** Differential Thermogravimetry, DTG results of the macromonomers and the polymer brushes.

Sample	Peak 1, °C	Peak 2, °C	Peak 3, °C
NBE-PLLA	267.51		
NBE-PCL		319.66	
NBE-(PLLA)_2_ #1	261.44		
NBE-(PLLA)_2_ #2	254.77		
NBE-(PCL)_2_ #1		326.94	
NBE-(PCL)_2_ #2		319.06	
DB-PLLA	266.90		440.95
DB-PCL		319.66	442.77
SDBC	277.82	331.18	442.77
BDBC	274.79	322.69	436.10
B-(PLLA-b-PCL-b-PLLA)	291.77	335.43	435.49

**Table 5 polymers-11-01606-t005:** Differential scanning calorimetry results of the macromonomers and the polymer brushes.

Sample	(Tm_1_)_PCL_, °C	(ΔHm_1_)_PCL_, J/g ^a^	(Tm_2_)_PLLA_, °C	(Tm_3_)_PLLA_, °C	(ΔHm)_PLLA_, J/g ^a^	(Tcc)_PLLA_, °C
NBE-PCL	53	85 (60.5%)				
NBE-(PCL)_2_ #1	51	80 (57.3%)				
NBE-PLLA				139	52 (38.4%)	82
NBE-(PLLA)_2_ #1			106	131	7 (5.0%)	
DB-PCL	46	62 (44.3%)				
DB-PLLA				138	12 (9%)	
BDBC	43	69 (49.4%)	104		5 (3.8%)	
B-(PLLA-b-PCL-b-PLLA)	48	59 (42.3%)	122	132	30 (22.0%)	

^a^ The crystallinity fraction is give in parenthesis.

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
