# Peer review of "Macromolecular Brushes Based on Poly(L-Lactide) and Poly(ε-Caprolactone) Single and Double Macromonomers via ROMP. Synthesis, Characterization and Thermal Properties"

_polymers, 2019, doi:10.3390/polym11101606_

Round 1

Reviewer 1 Report

The paper under consideration claims to report synthesis of complex macromolecular architectures via ring opening metathesis polymerization by using polycaprolactones and polylactides as macro monomers.  Different architectures are synthesized such as statistical and block copolymers. Furthermore, thermal gravimetric analysis of the synthesized architectures is compared.

However, It is not clearly defined what is novelty of current study. Authors themselves describe at numerous occasions that “ has been done according to our previously reported procedure” . Therefore, authors need to clearly explain what is novelty of this article compared to their own previous papers and other literature. Only thermal analysis of the synthesized polymers is not enough for publication in “Polymers”. Furthermore, thermal analysis is also limited to thermal degradation behavior. Dynamic scanning colorimetery analysis revealing variations in the Tg, Tm etc may reveal very useful information.

Therefore, I don’t think the article in its current form is suitable for publication in “Polymers”

Some minor comments for authors to consider are

Title is too long and should be modified Name and brand of columns should be included There are number of two-column, two row tables, that can be combined in a single comprehensive table – may be table 1-4 Authors wrote several time “statistical polymer” what is proof or argument that it is a statistical polymer not random polymer? What about giving TGA thermogram along with DTGA? DSC analysis can also add some important information regarding synthesized polymer brushes

Author Response

Reviewer #1

The paper under consideration claims to report synthesis of complex macromolecular architectures via ring opening metathesis polymerization by using polycaprolactones and polylactides as macro monomers.  Different architectures are synthesized such as statistical and block copolymers. Furthermore, thermal gravimetric analysis of the synthesized architectures is compared.

However, It is not clearly defined what is novelty of current study. Authors themselves describe at numerous occasions that “ has been done according to our previously reported procedure” . Therefore, authors need to clearly explain what is novelty of this article compared to their own previous papers and other literature. Only thermal analysis of the synthesized polymers is not enough for publication in “Polymers”. Furthermore, thermal analysis is also limited to thermal degradation behavior. Dynamic scanning colorimetery analysis revealing variations in the Tg, Tm etc may reveal very useful information.

Therefore, I don’t think the article in its current form is suitable for publication in “Polymers”

The major concern of the reviewer is the novelty of this work. It is true that in the past several papers have been published referring to the synthesis of graft copolymers via Ring Opening Metathesis Polymerization, ROMP and the use of the macromonomer technique. Fewer works reported the synthesis of polymer brushes by the homopolymerization of suitable macromonomers with the same polymerization technique. In our previous work, published in Macromolecules we described the synthesis of PCL and PLLA macromonomers and their subsequent use for the synthesis of statistical and block copolymacromonomers. In addition, we studied their thermal properties and in particular their crystallization behavior and how this is affected by the macromolecular architecture. The present study is a step forward in our ongoing project to control the synthesis of polymer brushes. We employed single PLLA and PCL macromonomers to synthesize a triblock brush by sequential addition of the macromonomers. The successful synthesis of the desired product revealed that ROMP is an efficient and very well controlled polymerization method for the synthesis of complex macromolecular architectures. We also synthesized double PLLA and PCL macromonomers with a norbornene polymerizable group. The ROMP of double macromonomers for the synthesis of double polymer brushes has been the subject of very few studies and in most of them the yield was far from quantitative and the molecular weight distribution rather broad. In the present study the yields are excellent and the control of the molecular characteristics in a very good level. Additionally, there is a possibility to synthesize statistical and block double copolymacromonomers leading to the formation of novel structures. This approach opens the route for the synthesis of a new family of products with brush structures. Along these lines recently we synthesized similar structures based on poly(styrene oxide) macromonomers.   

Some minor comments for authors to consider are

Title is too long and should be modified Name and brand of columns should be included There are number of two-column, two row tables, that can be combined in a single comprehensive table – may be table 1-4 Authors wrote several time “statistical polymer” what is proof or argument that it is a statistical polymer not random polymer? What about giving TGA thermogram along with DTGA? DSC analysis can also add some important information regarding synthesized polymer brushes 

The title of the manuscript was shortened as the reviewer suggested. Tables 3 and 4 were merged to one table. The term “statistical” is more general than the term “random”. Since we don’t know how random are the copolymers it is better to use the term “statistical”. TGA and DTG plots more or less provide the same information. In order to avoid having more Figures in the manuscript we decided to provide the more comprehensive DTG plots, since it is easier in this case to make comparisons between the samples and to identify their region of thermal decomposition. DSC data were added in the manuscript as the reviewer suggested.    

Reviewer 2 Report

Manuscript Title:

Complex Macromolecular Architectures via Ring Opening Metathesis Polymerization, ROMP, Based on Poly(L-Lactide) and Poly(ε-Caprolactone) Single and Double Macromonomers. Synthesis, Characterization and Thermal Properties

Manuscript ID : polymers-591882

Recommendation: Publish in Polymers after minor revisions

Comments:

In this manuscript, the authors presented a comprehensive study on the synthesis of various polymer brushes using PLLA and PCL single and double macromonomers. The structural information of the polymer brushes were determined by NMR and SEC. In addition, the thermal properties were also tested in this study, which confirmed that the thermal decomposition is not affected by the macromolecular architecture.

Besides, this manuscript is well-written, and well-organized. Thus, I suggest the acceptance by Polymers after addressing a few minor concerns.

What’s the difference between the method #1 and the method #2 for preparing NBE-(PLLA)2 and NBE-(PCL)2 (in Table1)?

In Line 111 & Line 129, the authors described how to use #2 to prepare NBE-(PCL)2 and NBE-(PLLA)2. But the authors didn’t mention how to use #1 to prepare those two macromonomers. Also, in Table 1, it is observed that the macromonomers prepared by #1 and #2 have different molecular characteristics. What caused this difference?

To be consistent with other parts in the manuscript, “NBE-(PLA)2” should be replaced by “NBE-(PLLA)2”; “NBE-PLA” should be replaced by “NBE-PLLA” in Table 1 and Figure 2;

Please check Table 1 and Figure 2

In Table 2, there was three footnotes “a”, “b”, “c” at the bottom of the table. But there was only “a” present in the table. The author needs to double check and edit it.

In Figure 5 and Figure 6, there was no X-axis name & there was no units for the X-axis values. In Figure 10, 12, 14, there was no units for X-axis values.

Author Response

Reviewer #2

In this manuscript, the authors presented a comprehensive study on the synthesis of various polymer brushes using PLLA and PCL single and double macromonomers. The structural information of the polymer brushes were determined by NMR and SEC. In addition, the thermal properties were also tested in this study, which confirmed that the thermal decomposition is not affected by the macromolecular architecture.

Besides, this manuscript is well-written, and well-organized. Thus, I suggest the acceptance by Polymers after addressing a few minor concerns.

What’s the difference between the method #1 and the method #2 for preparing NBE-(PLLA)2 and NBE-(PCL)2 (in Table1)?

In Line 111 & Line 129, the authors described how to use #2 to prepare NBE-(PCL)2 and NBE-(PLLA)2. But the authors didn’t mention how to use #1 to prepare those two macromonomers. Also, in Table 1, it is observed that the macromonomers prepared by #1 and #2 have different molecular characteristics. What caused this difference?

We are sorry for the misunderstanding. Samples NBE(PLLA)2 #1 and NBE(PLLA)2 #2 are two different double macromonomers of PLLA having different molecular weights. However, both samples were prepared by the same method, which is described in the Experimental Section. To avoid long texts we provided the quantities for the synthesis of sample #2 only. The same situation holds for the samples NBE(PCL)2 #1 and NBE(PCL)2 #2. They were prepared by the same methodology, as described in the manuscript. The symbols #1 and #2 are just used to differentiate samples with different molecular weights. This point was clarified in the text.

To be consistent with other parts in the manuscript, “NBE-(PLA)2” should be replaced by “NBE-(PLLA)2”; “NBE-PLA” should be replaced by “NBE-PLLA” in Table 1 and Figure 2;

Please check Table 1 and Figure 2

We are sorry for this mistake. The same terminology was employed throughout the text for the PLLA macromonomers.

In Table 2, there was three footnotes “a”, “b”, “c” at the bottom of the table. But there was only “a” present in the table. The author needs to double check and edit it

We are sorry for this mistake. The superscripts were added in Table 2.

In Figure 5 and Figure 6, there was no X-axis name & there was no units for the X-axis values. In Figure 10, 12, 14, there was no units for X-axis values.

The x-axis for these figures is time in minutes.

Reviewer 3 Report

This manuscript describes an interesting synthetic method to create PLA and PCL based brush polymers. The synthetic details including the synthesis of macromonomers and the subsequent brush polymers are well-written and easy to understand. However, I do have several comments:

Can you provide the degree of polymerization in the macromonomers and brush polymers? There might be too many figures and tables, I suggest putting some of them to the supporting information.

Author Response

Reviewer #3

This manuscript describes an interesting synthetic method to create PLA and PCL based brush polymers. The synthetic details including the synthesis of macromonomers and the subsequent brush polymers are well-written and easy to understand. However, I do have several comments:

Can you provide the degree of polymerization in the macromonomers and brush polymers? There might be too many figures and tables, I suggest putting some of them to the supporting information. 

The degrees of polymerization of the brushes were added in the Table with the molecular characteristics according to the reviewer’s suggestion. Tables 3 and 4 were merged to one Table and Figure 15 was moved to the Supporting Information Section.

Round 2

Reviewer 1 Report

adequately revised